# Antibacterial Activity of Banglene Extracted from Indonesian Ginger “Bangle” Against *Porphyromonas gingivalis*

**DOI:** 10.3390/ijms26051787

**Published:** 2025-02-20

**Authors:** Mayu Sebe, Satoka Senoura, Kiyoshi Miura, Wako Kobayashi, Nagisa Yano, Gaku Yamauchi, Kenichi Harada, Yoshiyasu Fukuyama, Miwa Kubo, Keiji Murakami

**Affiliations:** 1Department of Clinical Nutrition, Faculty of Health Science and Technology, Kawasaki University of Medical Welfare, 288 Matsushima, Kurashiki 701-0193, Japan; sebe@mw.kawasaki-m.ac.jp (M.S.);; 2Faculty of Pharmaceutical Sciences, Tokushima Bunri University, Tokushima 770-8514, Japan

**Keywords:** bangle, *Zingiber purpureum* Rosc, banglene, antibacterial activity, *Porphyromonas gingivalis*, periodontitis

## Abstract

Periodontitis is one of the most common diseases associated with the lifestyle habits of adults and is caused by the formation of biofilms, called dental plaques, in periodontal pockets by oral bacteria, such as *Porphyromonas gingivalis*. Bangle, *Zingiber purpureum* Rosc. (Indonesian ginger), a native Indonesian plant, has been traditionally consumed as food and medicine across Southeast Asia. The *cis*- and *trans*-banglenes, components of the rhizomes of *Z. purpureum*, have been reported to possess neurotrophic activity. Hexane extract of bangle exhibited antibacterial activity against *P. gingivalis*, with a minimum inhibitory concentration of 8 μg/mL. We isolated several compounds from the active fractions through the bioassay-guided isolation of hexane extract. Further, we found that *c*- and *t*-banglene inhibited the growth of *P. gingivalis* at 4 µg/mL; however, these compounds showed no antibacterial effects against oral microorganisms. We also observed that *c*- and *t*-banglenes resulted in 47% and 40% reductions in biofilm formation. In conclusion, our results suggest that banglene has specific antibacterial effects against the periodontopathogen *P. gingivalis*, with minimal impact on oral microorganisms. Thus, banglene has potential applications in the prevention of periodontitis without the risk of substituted microbisms.

## 1. Introduction

Periodontitis is one of the most common diseases associated with lifestyle habits in adults. It is a chronic inflammatory disease that destroys periodontal tissues, such as periodontal ligaments and alveolar bone, and is the cause of most cases of tooth loss [1]. Periodontitis is caused by the formation of biofilms, called dental plaques, in periodontal pockets by oral bacteria, such as *Porphyromonas gingivalis*. *P. gingivalis* is a Gram-negative oral anaerobic bacterium that can be detected at levels reaching 85% in periodontal pockets [2]. *P. gingivalis* is the keystone periodontal pathogen, a member of the group named “the red complex”, which has been strongly associated with advanced periodontal lesions, and is categorized together with *Treponema denticola* and *Tannerella forsythia* [3,4,5].

Recently, periodontitis has been associated with various systemic diseases, including Alzheimer’s disease, cardiovascular disease, diabetes, obesity, metabolic syndrome, and preterm birth [6]. Rhizomes of *Zingiber purpureum* Rosc. (synonyms: *Z. cassumunar* Roxb. and *Z. montanum* (Koenig) Link ex A. Dietr.), also called “bangle”, comprise a tropical ginger widely distributed in Southeast Asia [7]. Previous studies using the rhizomes of *Zingiber cassumunar* (a *Z. purpureum* synonym) have demonstrated that the isolated compounds exhibit various biological activities, such as anti-inflammatory and analgesic activities [8,9,10,11,12]. Used as a spice, this plant has applications in the traditional Indonesian medicine *Jamu* and has been used to treat fever, headache, stomach pain, rheumatism, and obesity, and as an ingredient in postpartum herbal medicine [13]. In our previous studies, *t*- [7] and *c*-banglenes [13] were successfully isolated from the methanol extract of the rhizomes of *Z. purpureum* and showed neurotrophic activity in PC12 cells [14,15,16]. Although it has a dark yellow color similar to that of common ginger, it does not contain gingerol or shogaol, which are typical components of ginger. However, it contains curcumin and phenylbutenoids, which are not found in ginger, and its constituents are different [17]. Bangle-rhizome extract tablets are commercially available in Japan. If we can scientifically prove that bangle is effective against periodontitis, this plant can not only be used to prevent periodontitis and reduce the risk of dementia but also be used as a useful food with the effect of improving cognitive function. In this study, we found for the first time that bangle has specific anti-bacterial activity against periodontitis-causing bacteria. Therefore, we searched for its active compounds and also investigated the effects of the active ingredients on biofilm.

## 2. Results

### 2.1. Antibacterial Effects of Z. purpureum Extracts on the Growth of P. gingivalis, and Isolation of Active Compounds

We found that the whole methanol extract of *Z*. *purpureum* exhibited antibacterial activity against *P. gingivalis* (minimum inhibitory concentration (MIC): 32 μg/mL). Therefore, we sequentially extracted *Z. purpureum* powder using various solvents (hexane, ethyl acetate, methanol, and water) to investigate the antibacterial activity of the extracts. The MIC values against *P. gingivalis* for the hexane, ethyl acetate, methanol, and aqueous extracts were 8, 64, 32, and 512, respectively (Table 1). These results indicate that the hexane extract exhibited the highest antibacterial activity.

To identify the substances with high antibacterial activity, we fractionated and purified the hexane extract and isolated several compounds. Compounds isolated from the hexane extract were identified using high resolution mass spectrometry (HRMS) and nuclear magnetic resonance (NMR) and confirmed, via comparison with the literature, as vanillin, linoleic acid, (*E*)-4-(3,4-dimethoxyphenyl)but-3-enyl acetate, (*E*)-1-(3,4-dimethoxyphenyl)buta-1,3-diene, (*E*)-4-(3,4-dimethoxyphenyl)but-3-en-1-ol, *trans*-banglene [7], *cis*-banglene [13], 3,4-bis(3,4-dimethoxyphenyl)-1,5-octadiene, (*cis*-1,2-bis[(*E*)-3,4-dimethoxystyryl]cyclobutane, and *cis*-3(2,4,5-trimethoxyphenyl)-4[(*E*)-3,4-dimethoxystyryl]cyclohexene (Figure 1). The MICs of these compounds against *P. gingivalis* are listed in Table 2. Among the identified compounds, both *c*- and *t*-banglenes exhibited the highest antibacterial activity (MIC: 4 μg/mL), respectively. Linoleic acid also showed strong antibacterial activity; however, its methyl ester form showed no activity. Since the antibacterial activity of the related fatty acids against *P. gingivalis* has already been reported [18], we focused on the antibacterial activity of *c*- and *t*-banglenes in this study.

### 2.2. Evaluation of the Antibacterial Effects of c- and t-Banglenes on the Growth of P. gingivalis

In our previous studies, *c*- and *t*-banglenes showed neurotrophic activity in PC12 cells [14,15,16]. Given this known activity relevant to improving cognitive function, we focused on the antibacterial effects as a novel function of banglene.

We investigated the antibacterial effects of *c*- and *t*-banglene on several strains of *P. gingivalis*. The MIC values of *c*- and *t*-banglenes against *P. gingivalis* are shown in Table 3. The *c*- and *t*-banglenes inhibited the growth of *P. gingivalis* W50 at 4 and 4 µg/mL, of *P. gingivalis* W83 at 2 and 4 µg/mL, of *P. gingivalis* 381 at 4 and 4 µg/mL, of *P. gingivalis* 1121 at 64 and 64 µg/mL, and of *P. gingivalis* TDC60 at 16 and 64 µg/mL, respectively. These results indicate that *c*- and *t*-banglenes exhibit antibacterial effects against most *P. gingivalis*, including the standard strains.

### 2.3. Antibacterial Effects of c- and t-Banglenes on the Growth of Oral Microorganisms

Next, we measured the antibacterial effects of *c*- and *t*-banglenes against various oral microorganisms, including Gram-positive and -negative bacteria and fungi. The *c*- and *t*-banglenes showed no antibacterial effects (MIC: >256 µg/mL) against *Fusobacterium nucleatum* and *Actinobacillus actinomycetemcomitans*, which are also periodontal pathogens. As shown in Table 4, *c*- and *t*-banglenes inhibited the growth of *Actinomyces viscosus* at 32 and 8 µg/mL. However, *c*- and *t*-banglenes showed no antibacterial effects against other pathogens, such as *Escherichia coli*, *Pseudomonas aeruginosa*, *Staphylococcus aureus*, *Streptococcus mutans*, *Streptococcus pyogenes*, and *Candida albicans*. Thus, our results suggest that banglene possesses antibacterial effects against only the periodontal pathogen-specific bacterium *P. gingivalis*, with minimal effects on other oral microorganisms.

### 2.4. Antibiofilm Effects of c- and t-Banglenes on the Growth of P. gingivalis

Dental biofilms that form in the oral cavity play a critical role in the pathogeneses of numerous infectious oral diseases, including periodontal diseases. Therefore, we investigated whether banglene exerts any antibiofilm effect, using a Cell Desk–biofilm forming assay with *P. gingivalis* ATCC33277. To confirm the inhibitory effects of biofilm forming, Banglene was added at a sub-MIC concentration (2 µg/mL), not affecting the bacterial growth.

A Cell Desk–biofilm formation assay at 48 h revealed that *c*-banglene significantly inhibited biofilm formation by *P. gingivalis* (Figure 2), resulting in a 47% reduction in the biofilm mass (Figure 2). We also showed that *t*-banglene significantly inhibited 48 h biofilm formation, resulting in a 40% reduction in the biofilm mass. These results revealed that *c*- and *t*-banglenes exhibit antibiofilm effects against *P. gingivalis*.

## 3. Discussion

In recent years, natural products derived from medicinal plants have received much attention as alternative medicines for preventing periodontitis, and many studies have been reported on this topic [19]. Several studies have shown that green tea catechins and their principal constituent, epigallocatechin-3-gallate (EGCG), are well known for their anti-bacterial effects against *P. gingivalis*. The MIC values of green tea extract and EGCG against *P. gingivalis* were reported to be from 125 to 1000 µg/mL [20]. Carrol et al. reported that *Pistacia lentiscus* fruits showed an MIC of 8 µg/mL, and *Zanthoxylum armatum* fruits/seeds an MIC of 16 µg/mL against *P. gingivalis* [21]. The MIC values of essential oil extracted from the gum of *Pistacia atlantica* subsp. *kurdica* were reported to be 12.5 µg/mL against *P. gingivalis* [22]. The MIC values of *trans*-cinnamaldehyde, a safe extract from natural plants, showed an MIC of 39.07 µg/mL against *P. gingivalis* [23]. Shirai et al. reported that mulberry extracts showed an MIC of 62.5–125 µg/mL against *P. gingivalis* [24]. Curcumin, extracted from the root of turmeric, demonstrated antibacterial activity against *P. gingivalis* at an MIC of 15 μg/mL [25]. We previously reported that the ethyl acetate extract of bangle contained curcumin, but the hexane extract did not [17]. In the current study, the antibacterial activity of the ethyl acetate extract of bangle was not strong against *P. gingivalis*, and it is assumed that the curcumin content in bangle would be low, suggesting that the antibacterial effect of bangle is due to compounds other than curcumin. The hexane extract of bangle exhibited antibacterial activity against *P. gingivalis* (MIC: 8 μg/mL). Bioassay-guided isolation of hexane extracts revealed that *c*- and *t*-banglenes both inhibited the growth of *P. gingivalis* at 4 µg/mL. Because the chemical composition of each extract was not exactly the same, it was difficult to determine the percentage of the chemical content of each extract. However, the *c*- and *t*-banglenes could be at their highest in the hexane layer [17].

Furthermore, biofilm formation of *P. gingivalis* was inhibited in the presence of 2 µg/mL of banglene, a concentration that does not affect the growth of *P. gingivalis*. During the initial phase of biofilm formation in *P. gingivalis*, surface structures such as fimbriae, lipopolysaccharides, and capsules play critical roles [26]. The mechanisms underlying extracellular matrix synthesis in the next phase of biofilm formation in *P. gingivalis* have not yet been clarified. In this study, we investigated the effects of banglene on gene expression during biofilm formation.

Banglene has *c*- and *t*- stereoisomers, which are present in Indonesian ginger in approximately the same ratio. No significant differences were observed between the *c*- and *t*-types in terms of antibacterial or antibiofilm activity against *P. gingivalis*.

As for the prevention of periodontitis, since periodontitis-related bacteria, such as *P. gingivalis*, are all endemic to the oral cavity, long-term use of disinfectants or antibacterial agents causes substituted microbism. However, since banglene has been shown to have almost no antibacterial activity against oral microorganisms, except for *P. gingivalis*, the risk of substituted microbes is expected to be very low, even with its long-term use.

This study showed that banglene, a constituent of *Z. purpureum*, exhibited antibacterial and antibiofilm activities against the periodontal pathogen *P. gingivalis* but not against other oral microorganisms. Regarding the species-specificity, novel zafirlukast derivatives have been reported to exhibit selective antibacterial activity against *P. gingivalis* [27]. Bacteriocins are peptides or proteins with antibacterial activity which are synthesized by bacteria; many bacteriocins have a relatively narrow spectrum [28]. Some bacteriocins have been reported to be specific to *P. gingivalis* and are thought to exert their effects through interaction with the cell wall or membrane-associated binding sites in *P. gingivalis* [29,30]. Bangle might be related to a specific protein from *P. gingivalis*. Murai et al. reported that curcumin suppressed the activity of some dipeptidyl peptidases, resulting in antibacterial activity against *P. gingivalis* [31]. Since banglene has a partially similar structure (feruloyl-type group) to curcumin, it is possible that it may have similar effects. However, in this study, some compounds were found to have feruloyl structures but weak activity, such as (*E*)-4-(3,4-dimethoxyphenyl)but-3-enyl acetate and (*E*)-4-(3,4-dimethoxyphenyl)but-3-en-1-ol. Therefore, it is possible that the stereochemical structure is also important for the expression of the activity. Several mechanisms could be responsible for the species-specific antibacterial activity of banglene. We plan to investigate the detailed mechanisms in the future.

Our results suggest that banglene has the potential to prevent periodontitis without the risk of a substituted microbism. Furthermore, the safety of bangle-rhizome extract has been evaluated through studies on its acute and chronic oral toxicity in rats, as well as a clinical trial in humans, with both confirming its safety [32]. Thus, it is a natural compound useful for the prevention of periodontitis.

Recently, periodontitis has been reported to be associated with various systemic diseases, including Alzheimer’s disease, cardiovascular disease, diabetes, obesity, metabolic syndrome, and preterm birth [6,33]. Alzheimer’s disease is a gradually progressive form of dementia that begins in old age and is primarily characterized by cognitive dysfunction, particularly memory impairment [34]. Periodontal tissues of patients with periodontitis show increased production of inflammatory cytokines and elevated levels of inflammatory markers such as C-reactive protein, tumor necrosis factor-alpha, and interleukin-6 in the blood [35]. Periodontal bacteria can invade the brain and elicit an inflammatory response. Inflammation of the periodontal tissue and periodontopathogenic bacteria may directly or indirectly spill over into the brain and exacerbate the pathophysiology of Alzheimer’s disease [36]. The possibility that banglene may be effective against periodontal diseases suggests that in addition to its direct effect on brain neurons in dementia, it may indirectly contribute to the prevention of dementia through the prevention of periodontal diseases.

## 4. Material and Methods

### 4.1. Plant, Extraction, and Isolation

General Experiment. Silica gel column chromatography (CC) was carried out on KANTO CHEMICAL silica gel 60 N, Wako C-300, and Kiselgel 60 (70–230 mesh and 230–400). HPLC was performed on a JASCO PU-1580 equipped with a JASCO UV-1575 detector (Tokyo, Japan). The NMR experiments were performed on a Varian Unity 600 MHz (Santa Clara, CA, USA) or Bruker 500 MHz (Billerica, MA, USA) NMR Spectrometer. ^1^H spectra were referenced to the deuterated solvent peaks and TMS was used as internal standard.

Plant Material. Rhizomes of *Zingiber purpureum* Rosc. were purchased from an Indonesian market by Hosoda Co., Ltd. (Fukui, Japan) in April 2013 and identified by Dr. E. Kato (Hosoda Co., Ltd. company). A voucher specimen (1794, 1799) has been deposited in the Institute of Pharmacognosy, Tokushima Bunri University, Japan.

Extraction and Isolation. The rhizomes of *Z. purpureum* were powdered (103.4 g) and successively extracted by solvents of increasing polarity (hexane, ethyl acetate, methanol, and water), with 2 × 300 mL of each solvent being placed under sonication for 1 h. Each extract was concentrated under vacuum and dried under a nitrogen stream until the complete evaporation of the solvent. The extractions yielded extracts of 3.3 g of hexane (3.2%), 5.7 g of ethyl acetate (5.5%), and 7.2 g of methanol (7.0%), as well as 8.8 g of water (8.5%). In addition, the powder was extracted separately with methanol (called the “full methanol extract”).

The hexane extract (3.3 g) was chromatographed on a Si gel column eluted with Hexane–EtOAc (4:1) to give fractions 1–19, and fr. 5, fr. 10, and fr. 16 were determined to be (*E*)-1-(3,4-dimethoxyphenyl)buta-1,3-diene (0.20 g) [37], (*E*)-4-(3,4-dimethoxyphenyl)but-3-enyl acetate (0.30 g) [38], and (*E*)-4-(3,4-dimethoxyphenyl)but-3-en-1-ol (0.30 g) [39]. Fraction 13 (0.10 g) was subjected to Si gel chromatography eluting with Hexane–Ether (3:2) to give fractions 20–24. Fraction 22 (64.8 mg) was purified by HPLC (Cosmosil SL-II, *ϕ*10 × 250 mm) using CH_2_Cl_2_–Ether (95:5) to give (*cis*-1,2-bis[(*E*)-3,4-dimethoxystyryl]cyclobutane (1.5 mg) [40], *t*-banglene (6.1 mg) [7,41], *c*-banglene (2.3 mg) [13,41], vanillin (1.0 mg) [42], and 3,4-bis(3,4-dimethoxyphenyl)-1,5-octadiene (1.0 mg) [43]. Fraction 14 (0.20 g) was subjected to Si gel chromatography eluting with CH_2_Cl_2_–Ether (20:1) to give fractions 25–29; fr. 27 was *t*-banglene (7.3 mg). Fraction 25 (0.10 g) was chromatographed on a Si gel column eluted with Hexane–EtOAc (7:3) to give fractions 30–34. Fraction 31 was subjected to Si gel chromatography eluting with Hexane–Ether (1:1) to give *cis*-3(2,4,5-trimethoxyphenyl)-4[(*E*)-3,4-dimethoxystyryl]cyclohexene (8.5 mg) [40]. Linoleic acid was isolated from the hexane extract, identified via HRMS and NMR and compared with a commercially available compound (Nacalai tesque code 20513-41, Kyoto, Japan); linoleic acid methyl ester was prepared via methylation of linoleic acid.

### 4.2. Bacterial and Fungal Strains and Growth Conditions

The bacterial and fungal strains used in this study are listed in Table 5. *E. coli*, *P. aeruginosa* and *Staphylococcus* spp. were grown in Muller–Hinton broth (Becton Dickinson, Sparks, MD, USA) supplemented with 50 µg/mL CaCl_2_ and 25 µg/mL MgCl_2_. *A. actinomycetemcomitans*, *Actinomyces* spp., and *Streptococcus* spp. were grown anaerobically during brain–heart infusion broth (Becton Dickinson). *F. nucleatum* and *P. gingivalis* strains were grown anaerobically in brain–heart infusion broth supplemented with 5 µg/mL hemin and 1 µg/mL menadione. *Candida* spp. were grown in potato dextrose broth. For the susceptibility assay, RPMI broth (Thermo Fisher Scientific, Waltham, MA, USA) was used for *Candida* spp.

### 4.3. Susceptibility Assay

The MIC was determined using the microbroth dilution method. As there is no standard method described in the CLSI (Clinical and Laboratory Standards Institute, Berwyn, PA, USA) guidelines for culturing and evaluating the MIC of *P. gingivalis*, we followed previously described methods [15,16]. Approximately 10^6^ colony-forming units (CFU)/mL of each bacterial culture were inoculated into 100 μL of medium (two-fold serial dilution) in 96-well plates (AS ONE, Osaka, Japan) and incubated, either anaerobically (for *A. actinomycetemcomitans*, *Actinomyces* spp., *Streptococcus* spp., *F. nucleatum*, and *P. gingivalis*) or aerobically (for *E. coli*, *P. aeruginosa*, *Staphylococcus* spp., and *Candida* spp.), at 37 °C for 24 or 48 h. After the incubation, bacterial growth was recorded visually. The MIC values were defined as the lowest concentration associated with a lack of visible growth in the well, compared with control.

### 4.4. Biofilm Formation Assay

For the biofilm formation assays, brain–heart infusion broth supplemented with 5 µg/mL hemin and 1 µg/mL menadione was used for *P. gingivalis* ATCC33277. A biofilm was formed on the surface of a type I collagen-coated coverslip (Cell Desk LF1; Sumitomo Bakelite, Tokyo, Japan) placed in a 24-well plate (AS ONE). A 40 μL (10^6^ CFU/mL) sample of *P. gingivalis* was transferred into a 24-well plate from the primary suspensions of the broth. The *c*- or *t*-banglenes were then added to a final concentration of 2 µg/mL, which is half the MIC for *P. gingivalis* ATCC33277 (4 µg/mL). This concentration allowed us to assess biofilm formation without complete bacterial growth inhibition. Broth alone was used as a control. The bacterial suspensions were incubated anaerobically at 37 °C for 48 h, using a method previously reported [44].

A crystal violet biofilm assay was performed to quantify the biofilm mass. After incubation, the coverslips were rinsed twice with distilled water, without disturbing the adherent biofilm. They were then stained with 500 μL of 0.1% crystal violet at 25 °C for 10 min. Excess staining was removed by gentle washing with distilled water. After drying, the stained biofilms were extracted from each well by adding 500 μL of ethanol, and the absorbance of the extract from the stained biofilm was measured at 570 nm using a microplate reader (model 357; Thermo Fisher Scientific).

### 4.5. Statistical Analysis

Statistical analyses were performed using JMP software version 13 (SAS Institute, Tokyo, Japan). A *p* value < 0.05 was considered statistically significant. The results are presented as the mean ± standard deviation. Multiple comparisons were performed using one-way analysis of variance, followed by the Tukey–Kramer honest significant difference test.

## 5. Conclusions

This study revealed that banglene exhibits significant antibacterial and antibiofilm activities against the periodontal disease-related bacterium *P. gingivalis*. However, it did not demonstrate antibacterial effects against other resident oral bacteria. With minimal impact on oral commensal bacteria, banglene poses a low risk of substituted microbisms, thus indicating its potential usefulness in preventing periodontal diseases. Given its commercial availability and high safety profile, banglene is expected to be utilized as a beneficial food product. Moving forward, further investigations are needed to elucidate the specific mechanisms by which banglene exerts its selective effects against *P. gingivalis*. Additionally, the anti-inflammatory effects of banglene on the host tissue (gingiva epithelium) will be explored to better understand banglene’s potential role in the prevention of periodontal diseases.

## Figures and Tables

**Figure 1 ijms-26-01787-f001:**
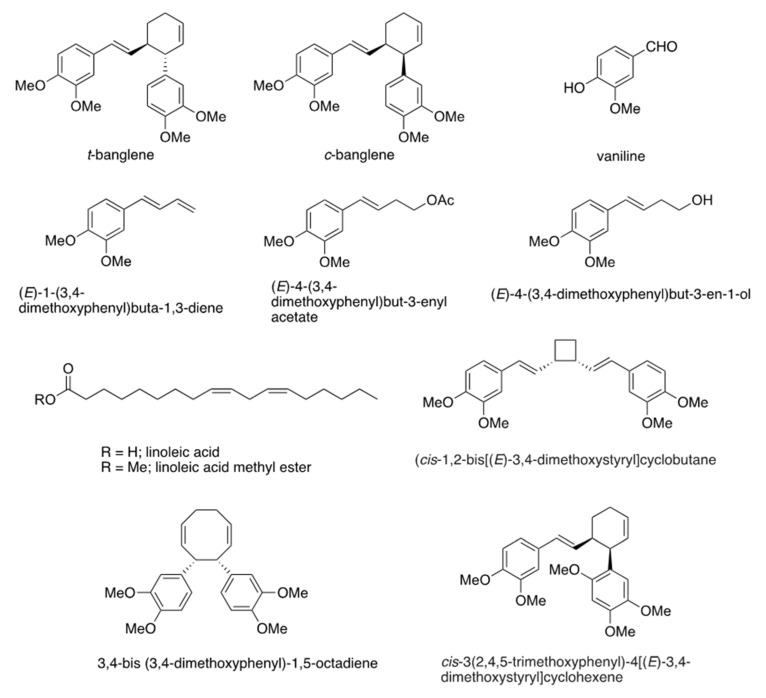
Structures of the compounds isolated from the hexane extract of *Z. purpureum*.

**Figure 2 ijms-26-01787-f002:**
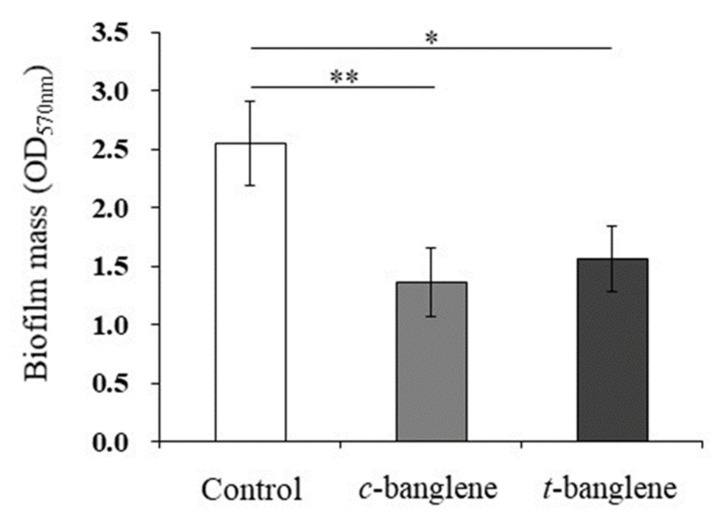
Antibiofilm effects of *c*-banglene and *t*-banglene on *P. gingivalis* ATCC 33277 via Cell Desk–biofilm forming assay. Brain–heart infusion broth supplemented with 5 µg/mL hemin and 1 µg/mL menadione was used. The *c*- or *t*-banglene was then added to a final concentration of 2 µg/mL. For the control, only the broth was used. Bacterial suspensions were incubated anaerobically at 37 °C for 48 h. ** Significant differences between the indicated groups at *p* < 0.01. * Significant differences between the indicated groups at *p* < 0.05 using a Tukey–Kramer honestly significant difference test (n = 3).

**Table 1 ijms-26-01787-t001:** Minimum inhibitory concentration (MIC) of *Zingiber purpureum* powder extract against *Porphyromonas gingivalis* ATCC 33277.

Sample	MIC (µg/mL)
Hexane extract	8
Ethyl acetate extract	64
Methanol extract	32
Aqueous extract	512
Full methanol extract	32

**Table 2 ijms-26-01787-t002:** MICs of compounds from the hexane extract against *P. gingivalis* ATCC 33277.

Sample	MIC (µg/mL)
*t*-banglene	4
*c*-banglene	4
vanillin	32
(*E*)-1-(3,4-dimethoxyphenyl) buta-1,3-diene	32
(*E*)-4-(3,4-dimethoxyphenyl) but-3-enyl acetate	256
(*E*)-4-(3,4-dimethoxyphenyl) but-3-en-1-ol	>1024
linoleic acid	4
(*cis*-1,2-bis[(*E*)-3,4-dimethoxystyryl] cyclobutane	32
3,4-bis(3,4-dimethoxyphenyl)-1,5-octadiene	256
*cis*-3(2,4,5-trimethoxyphenyl)-4[(*E*)-3,4-dimethoxystyryl] cyclohexene	8

**Table 3 ijms-26-01787-t003:** MICs of *c*-banglene and *t*-banglene against *P. gingivalis*.

Strain	MIC (µg/mL)	
	*c*-Banglene	*t*-Banglene
*Porphyromonas gingivalis* W50	4	4
*Porphyromonas gingivalis* W83	2	4
*Porphyromonas gingivalis* 381	4	4
*Porphyromonas gingivalis* 1121	64	64
*Porphyromonas gingivalis* TDC60	16	64

**Table 4 ijms-26-01787-t004:** MICs of *c*-banglene and *t*-banglene against oral microorganisms.

Strain	MIC (µg/mL)	
	*c*-Banglene	*t*-Banglene
*Fusobacterium nucleatum* JCM8532	>256	>256
*Escherichia coli* NBRC3972	>256	>256
*Pseudomonas aeruginosa* PAO1	>256	>256
*Actinobacillus actinomycetemcomitans* JCM2434	>256	>256
*Actinomyces israelii* ATCC10102	>256	>256
*Actinomyces naeslund* JCM8349	>256	>256
*Actinomyces viscosus* JCM8353	32	8
*Staphylococcus aureus* NBRC12732	>256	>256
*Staphylococcus epidermidis* 3762	>256	>256
*Streptococcus agalactiae* TUH01	>256	>256
*Streptococcus anginosus* NCTC10713	>256	>256
*Streptococcus constellatus* TW4496	>256	>256
*Streptococcus gordonii* ATCC10558	>256	>256
*Streptococcus intermedius* UNS46	>256	>256
*Streptococcus mitis* JCM12971	>256	>256
*Streptococcus oralis* ATCC10557	>256	>256
*Streptococcus mutans* MT8148	>256	>256
*Streptococcus mutans* UA159	>256	>256
*Streptococcus sobrinus* B13	>256	>256
*Streptococcus pyogenes* HOK1	>256	>256
*Candida albicans* CAD1	>256	>256
*Candida glabrata* JCM3761	>256	>256
*Candida tropicalis* JCM1541	>256	>256

**Table 5 ijms-26-01787-t005:** Bacterial and fungal strains.

Strain
*Actinobacillus actinomycetemcomitans* JCM2434
*Actinomyces israelii* ATCC10102
*Actinomyces naeslund* JCM8349
*Actinomyces viscosus* JCM8353
*Escherichia coli* NBRC3972
*Fusobacterium nucleatum* JCM8532
*Porphyromonas gingivalis* ATCC33277
*Porphyromonas gingivalis* TDC60
*Porphyromonas gingivalis* W50
*Porphyromonas gingivalis* W83
*Porphyromonas gingivalis* 1121
*Porphyromonas gingivalis* 381
*Pseudomonas aeruginosa* PAO1
*Staphylococcus aureus* NBRC12732
*Staphylococcus epidermidis* 3762
*Streptococcus agalactiae* TUH01
*Streptococcus anginosus* NCTC10713
*Streptococcus constellatus* TW4496
*Streptococcus gordonii* ATCC10558
*Streptococcus intermedius* UNS46
*Streptococcus mitis* JCM12971
*Streptococcus mutans* MT8148
*Streptococcus S. mutans* UA159
*Streptococcus oralis* ATCC10557
*Streptococcus pyogeues* HOK1
*Streptococcus sobrinus* B13
*Candida albicans* CAD1
*Candida glabrata* JCM3761
*Candida tropicalis* JCM1541

## Data Availability

The data presented in this study are available upon request from the corresponding author.

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
