# Peer review of "Antibacterial Activity of Banglene Extracted from Indonesian Ginger “Bangle” Against Porphyromonas gingivalis"

_ijms, 2025, doi:10.3390/ijms26051787_

Round 1

Reviewer 1 Report (Previous Reviewer 2)

Comments and Suggestions for Authors

The study is relevant considering the therapeutic properties of medicinal plants. Some considerations to the Authors:

Introduction:

- Provide a better description of "Bangle", its active components and medicinal properties. Add references, if necessary

- Expand the Introduction, citing studies that investigated the effects of these extracts against periodontopathogenic bacteria.

- Reinforce the justification for the study, considering the lack in the literature on the subject.

Discussion:

- Provide a better explanation of the probable mechanisms by which banglene exerts an antibacterial effect specifically against P. gingivalis, considering the literature.

- Add a brief discussion about the safety-toxicity of banglene.

Material and Methods:

4.1. Plant, Extraction, and Isolation: Mention the extraction and isolation protocol, adding the reference.

4.4. Biofilm Formation Assay: Mention how the concentrations of c- or t-Banglenes (2 μg/mL) were defined and add references, if necessary.

Conclusion:

- The conclusion supports the results of the study. Add perspectives for future research.

Author Response

Introduction

Comments 1: Provide a better description of "Bangle", its active components and medicinal properties. Add references, if necessary.

Comments 2: Expand the Introduction, citing studies that investigated the effects of these extracts against periodontopathogenic bacteria.

Comments 3: Reinforce the justification for the study, considering the lack in the literature on the subject.

Response 1-3: We appreciate your insight; however, to the best of our knowledge, this study is the first to report the effects of Bangle on oral bacteria, including periodontopathogenic species. Unfortunately, no prior studies specifically addressing this topic are available for citation at this time. Nonetheless, we have carefully revised the Introduction to better contextualize our study and highlight its novelty by citing related works on the active components of Bangle and its medicinal properties (Lines 43-46).

Discussion

Comments 4: Provide a better explanation of the probable mechanisms by which banglene exerts an antibacterial effect specifically against P. gingivalis, considering the literature.

Response 4: We have included a discussion on this topic in Lines 180-187. Specifically, we hypothesize that banglene may target a specific protein in P. gingivalis, similar to the mechanism reported by Murai et al. for curcumin. Their study demonstrated that curcumin suppresses the activity of certain dipeptidyl peptidases, resulting in antibacterial activity against P. gingivalis. Since banglene shares a partial feruloyl-type structure with curcumin, it is possible that it may exert similar effects. While our study has not yet confirmed the exact mechanisms, we plan to conduct further investigations to elucidate whether banglene indeed acts through a mechanism similar to curcumin.

Comments 5: Add a brief discussion about the safety-toxicity of banglene.

Response 5: In the revised manuscript, we have included a discussion on this topic (Lines 189-192). Kato et al. reported the safety assessment of bangle rhizome extract (BRE) on its acute and chronic oral toxicity in rats and clinical trial in humans [32]. According to the reference, the safety of BRE has been thoroughly evaluated. Acute toxicity tests in rats (up to 2,000 mg/kg/day), subacute toxicity tests (up to 1,000 mg/kg/day for 90 days), and human safety trials in healthy individuals (up to 850 mg/person/day for 28 days) have been conducted, with no abnormalities observed in any of these studies. Furthermore, the pharmacokinetics of BRE in healthy individuals following the oral administration of BRE tablets have been elucidated. Both trans-banglene and cis-banglene reached their maximum plasma concentrations approximately 1.8 hours post-administration, with peak plasma concentrations of 22.61 ± 6.70 ng/mL and 17.73 ± 5.61 ng/mL, respectively. These findings indicate a minimal risk of accumulation toxicity.

Material and Methods

Comments 6: 4.1. Plant, Extraction, and Isolation: Mention the extraction and isolation protocol, adding the reference.

Response 6: We have included the extraction and isolation protocol on this topic in Lines 209-242.

Comments 7: 4.4. Biofilm Formation Assay: Mention how the concentrations of c- or t-Banglenes (2 μg/mL) were defined and add references, if necessary.

Response 7: Our findings indicated that the MIC of Banglens for P. gingivalis ATCC33277 was 4 µg/mL, which completely inhibits its growth. In this study, we would like to focus on the inhibitory effects of Banglens on biofilm forming process. To determine an appropriate concentration below the MIC, we conducted preliminary experiments and selected a concentration of 2 µg/mL. This concentration ensures that P. gingivalis could continue to grow, allowing us to observe the initial attachment or biofilm maturation processes under conditions where bacterial proliferation was maintained. This explanation is provided in Lines 271–273.    As a reference for determining the concentration, we utilized the report by Sekita et al.

Sekita Y, Murakami K, Yumoto H, Mizuguchi H, Amoh T, Ogino S, Matsuo T, Miyake Y, Fukui H, Kashiwada Y. Anti-bacterial and anti-inflammatory effects of ethanol extract from Houttuynia cordata poultice. Biosci Biotechnol Biochem. 2016, 80, 1205-13.

Conclusion

Comments 8: The conclusion supports the results of the study. Add perspectives for future research.

Response 8: We have included the perspectives for future research Lines 296-300.

Reviewer 2 Report (New Reviewer)

Comments and Suggestions for Authors

In this manuscript by Sebe et al., the author has extracted several compounds from Indonesian Ginger “Bangle” and tested their activity against the pathogen Porphyromonas gingivalis. I have several issues, which the author need to address.

1. In the introduction the author should discuss more on the ginger, its medicinal values, any patents and clinical compounds from it. And also the importance of the pathogen.

2. In materials and methods, the author should separate the extraction and isolation procedure, and describe all, extraction process, HPLC, NMR etc in details separately along with the name and instrument specifications.

3.In the title, the author mentioned they tested against one pathogen, but in the materials and methods, they mentioned several strains why?

4. The author should consider modifying the title as per their experiments.

5. The author should provide the details of positive and negative controls used in the experiment.

6. Has the author used any standard antibiotics and antifungal agent in the experiment?

7.More experiments on the mechanism of antimicrobial action is needed.

8. Author should make a figure of the mode of action.

9. Conclusion is too short, needs modification, highlighting the importance of the work.

10.HPLC, NMR spectral images should be provided.

11. The author should compare their results with more recent references.

Comments on the Quality of English Language

moderate

Author Response

Comments 1: In the introduction the author should discuss more on the ginger, its medicinal values, any patents and clinical compounds from it. And also the importance of the pathogen.

Response 1: We appreciate your insight; we have carefully revised the Introduction to better contextualize our study and highlight its novelty by citing related works on the active components of Bangle and its medicinal properties (Lines 43-46).

Comments 2: In materials and methods, the author should separate the extraction and isolation procedure, and describe all, extraction process, HPLC, NMR etc in details separately along with the name and instrument specifications.

Response 2: We have included the extraction and isolation protocol on this topic in Lines 209-242.

Comments 3: In the title, the author mentioned they tested against one pathogen, but in the materials and methods, they mentioned several strains why?

Comments 4: The author should consider modifying the title as per their experiments.

Response 3-4: We examined the effects of Banglene on oral microorganisms to evaluate its broader antibacterial potential. However, we selected the current title to emphasize its specific effectiveness against P. gingivalis, which we consider the most important finding of our study.  To address any potential misunderstanding, we have revised the title of Table 4 to clarify its content and ensure it aligns with the scope and focus of the manuscript.

Comments 5: The author should provide the details of positive and negative controls used in the experiment.

Comments 6: Has the author used any standard antibiotics and antifungal agent in the experiment?

Response 5-6: In this work, we used DMSO as a negative control. This concentration of DMSO does not have antibacterial ability. For positive control, tetracycline was employed. Under our experimental conditions, the MIC of tetracycline was confirmed to be 0.25 µg/mL against P. gingivalis. Additionally, we tested another natural compound, Magnolia bark extract, as part of our comparative analysis. The MIC of Magnolia bark extract was confirmed to be 64 µg/mL against P. gingivalis.

Comments 7: More experiments on the mechanism of antimicrobial action is needed.

Response 7: We have included a discussion on this topic in Lines 180-187. Specifically, we hypothesize that banglene may target a specific protein in P. gingivalis, similar to the mechanism reported by Murai et al. for curcumin. Their study demonstrated that curcumin suppresses the activity of certain dipeptidyl peptidases, resulting in antibacterial activity against P. gingivalis. Since banglene shares a partial feruloyl-type structure with curcumin, it is possible that it may exert similar effects.   While our study has not yet confirmed the exact mechanisms, we plan to conduct further investigations to elucidate whether banglene indeed acts through a mechanism similar to curcumin.

Comments 8: Author should make a figure of the mode of action.

Response 8: As the exact mechanisms underlying the effects of banglene have not been fully elucidated in this study, it is challenging to create a figure of the mode of action. We plan to investigate the specific mechanisms in future studies, which may allow us to present a more comprehensive representation of the mode of action.

Comments 9: Conclusion is too short, needs modification, highlighting the importance of the work.

Response 9: We have included the perspectives for future research Lines 296-300.

Comments 10: HPLC, NMR spectral images should be provided.

Response 10: We provide NMR spectral images as supplemental material for your reference.

Comments 11: The author should compare their results with more recent references.

Response 11: In response, we have incorporated two additional recent studies on the antibacterial activity of natural compounds against P. gingivalis in Lines 142-145. These references provide further context and underscore the significance of our findings in relation to existing literature.

Round 2

Reviewer 1 Report (Previous Reviewer 2)

Comments and Suggestions for Authors

The adjustments improved the manuscript.

Author Response

Thank you for your kind feedback. We are glad to hear that the adjustments improved the manuscript. We greatly appreciate your valuable comments and suggestions, which were instrumental in enhancing the quality of our work.

Reviewer 2 Report (New Reviewer)

Comments and Suggestions for Authors

The author has made a few revisions, however, an essential part is missing, which is the study of the mode of antibacterial action. I feel without it, the current study is incomplete.

Comments on the Quality of English Language

ok

Author Response

Comments 1: The author has made a few revisions, however, an essential part is missing, which is the study of the mode of antibacterial action. I feel without it, the current study is incomplete.

Response1: 

We understand that elucidating the antibacterial mechanisms of banglene is crucial; however, this is a highly challenging task. Although numerous studies have reported the antimicrobial activity of natural extracts from plants, animals, insects, and microorganisms, in most cases, the specific antimicrobial compounds have not been identified. Furthermore, the mode of antibacterial action remains unclear for the majority of these compounds.

The most widely recognized mechanism of action involves disrupting the structure of the bacterial cell membrane. However, various hypotheses, such as the carpet-like pore model, barrel-stave pore model, and ring-like pore model, have been proposed to explain membrane disruption. Since banglene exhibits a highly selective antibacterial spectrum, its target is likely to be something other than the bacterial cell membrane. Given that banglene shares a similar partial structure with curcumin, it is possible that, like curcumin, it suppresses the activity of certain dipeptidyl peptidases. However, extensive experimental data will be required to confirm this hypothesis. We plan to investigate these antibacterial mechanisms in future research (Lines 187-189).

In this study, we are the first to demonstrate that banglene exhibits significant antibacterial and antibiofilm activities against P. gingivalis. We believe that our findings provide valuable insights and that this study should be of particular interest to the readers of this journal.

Comment for quality of English Langage : The English could be improved to more clearly express the research.

Response : Thank you for your feedback. To ensure the clarity and accuracy of our manuscript, we have had the English professionally edited by Editage. We have attached the certificate of editing for your reference.

Round 3

Reviewer 2 Report (New Reviewer)

Comments and Suggestions for Authors

Ok

This manuscript is a resubmission of an earlier submission. The following is a list of the peer review reports and author responses from that submission.

Round 1

Reviewer 1 Report

Comments and Suggestions for Authors

The manuscript entitled "Antibacterial activity of Banglene axtracted from Indonesian ginger “Bangle” against Porphyromonas gingivalis" by Mayu Sebe et al. reported susceptibility data for components of Zingiber purpureum  against Porphyromonas gingivalis.

Unfortunately, even if scientific issues addressed in the text are of interest, the manuscript is not suitable for publication on IJMS since several modifications are necessary before it can be considered for publications.

First of all due to the small number of P. gingivalis used. It is unclear whether the bacteria studied are of clinical origin. If so, please specify. It seems, the study was only done on one strain of P. gingivalis, perhaps ATCC? It is hard to believe that the efficacy of banglene reported is the trend in the whole bacteria; hence, the conclusions are weak compared with the aims of the work.

Second, the reviewer does not understand why the authors did not correlate the obtained results with a positive and negative controls (what are the guidelines for periodontitis?). Additionally, few work limitations are reported.

Periodontitis is an inflammatory condition caused due to microbial origin in the oral cavity. A variety of microorganisms were identified as disease-causing microbes. P.gingivalis is one of the periopathogen. Why banglene is efficacy only on P.gingivalis

Are the authors sure that the MIC results and in µg/ml?

Lines 51-63 there are repetitions. Please, to review.

Table 1 is not clear in the Results. Why did the authors use different extraction methods? Furthermore, it is necessary to list the percentages of the components of Z. purpureum and discuss the reason for the MIC=8 µg/ml result for Hexana extract compared to the other extracts.

In addition, the authors should also take into consideration a clinical strain in the antibiofilm effects of banglene.

Finally, regarding Material and Methods section, the reviewer does not understand why the authors don't mention guidelines to determine the susceptibility pattern. This even if there is no standardized method for non-drugs.

Comments on the Quality of English Language

Minor editing of English language required

Author Response

 We would like to thank the reviewers for their helpful comments and hope that we have now produced a more balanced and better account of our work. We believe that the revised manuscript is acceptable for publication in International Journal of Molecular Sciences.

Reviewer 2 Report

Comments and Suggestions for Authors

This is a relevant study that investigated the antimicrobial action of bioactive compounds from rhizome of Zingiber purpureum Rosc., especially banglenes, on Porphyromonas gingivalis and other oral microorganisms. Some considerations:

1) Add sufficiently clear and more detailed information about cell culture procedures. Please clarify in the text how the concentrations of the treatment extracts were selected/defined by the Authors;

2) The discussion of results is limited and insufficient and needs to be improved. Considering data available in the literature, a broad approach and discussion is needed on the isolation and purification methods of bioactive compounds from the rhizomes of Zingiber purpureum Rosc., culture conditions, characterization and properties of banglenes, factors and potential mechanisms that may be related to the its antimicrobial activity, specificity of action on certain species, pharmacokinetic and pharmacodynamic aspects, including bioavailability, safety and toxicity.

Author Response

(The authors gave the same response as above.)

Round 2

Reviewer 1 Report

Comments and Suggestions for Authors

The manuscript entitled "Antibacterial activity of Banglene axtracted from Indonesian ginger “Bangle” against Porphyromonas gingivalis" by Mayu Sebe et al. unfortunately is not suitable for publication on IJMS. As already highlighted in the previous review, several modifications are necessary before it can be considered for publications. The authors responded to the reviewer but did not edit the manuscript. The doubts expressed about the first manuscript should be explained in the second version

The study was only done on one strain of P. gingivalis ATCC. It is hard to believe that the efficacy of banglene reported is the trend in the whole bacteria; hence, the conclusions are weak compared with the aims of the work. The study should also be conducted on T denticola and T. forsythia. Please see: J Infect Dev Ctries 2021;15(11):1685-1693. doi:10.3855/jidc.14904

The authors should compare the results obtained not only with positive and negative controls, but also with at least one reference drug. Why banglene is efficacy only on P.gingivalis? Clearly discuss the results in the discussion Are the authors sure that the MIC results and in µg/ml?

Discuss the reason for the MIC=8 µg/ml result for Hexana extract compared to the other extracts.

In Material and Methods section, the authors must indicate the concentrations used to define the MICs and the method for preparing the inoculum

Author Response

We would like to thank the editor and reviewers for their helpful comments and hope that we have now produced a more balanced and better account of our work. We believe that the revised manuscript is acceptable for publication in International Journal of Molecular Sciences.

Reviewer 2 Report

Comments and Suggestions for Authors

The issues previously commented could be better addressed and discussed by the Authors in the manuscript.

Author Response

(The authors gave the same response as above.)

Round 3

Reviewer 2 Report

Comments and Suggestions for Authors

Considering the previous comments, some issues should be better addressed and discussed by the Authors. I suggest that the Authors complement the manuscript.

Author Response

Thank you for your valuable feedback on our manuscript.